# Predictors of fatal neurological complications among admitted COVID-19 patients with their implication in outcome: A Case Control study

**Javaria Aslam** [1,2☺‡*], **Shoaib Luqman** [3☺], **Sadaf Nazly** [2☺], **Alina Saeed** [2☺], **Muhammad Sohail Tariq** [1‡], **Sultan Yahya Mohammad Alfaife** [4‡], **Irrum Aneela** [5‡]

1 Department of Medicine, Qauid e Azam Medical College, Bahawalpur, Pakistan, 2 Department of Medicine, Sir, Sadiq Abbasi Hospital, Qauid e Azam Medical College, Bahawalpur, Pakistan, 3 Department of Neurology, Qauid e Azam Medical College, Bahawalpur, Pakistan, 4 Department of General Medicine, General Directorate of Health Affairs, Aseer Region, Ministry of Health, Abha, Saudi Arabia, 5 Department of Rehabilitation Medicine, Astley Ainslie Hospital, Edinburg, Scotland, United Kingdom

☺ These authors contributed equally to this work.
‡ JA, MST, SYMA and IA also contributed equally to this work.
* Javaslam50@gmail.com

**Data Availability Statement:** All relevant data are within the paper and its Supporting Information files.

## Abstract

### Background

COVID-19 is known to be associated to potentially fatal neurological complications; therefore, it is essential to understand the risk factors for its development and the impact they have on the outcome of COVID-19 patients.

### Aims

To determine the risk factors for developing fatal neurological complications and their outcome in hospitalized COVID-19 patients.

### Material and methods

Case control study based on hospitalized patients was conducted from July 15th 2021 to December 15th 2021. Cases and controls were COVID-19 confirmed patients with and without severe neurological manifestations. Age, comorbid conditions, vaccination status, Blood Sugar Random (BSR), D-dimers levels, anticoagulation type and dosage were taken as predictors (exposure variables) for developing neurological complications. In the case-only (subgroup) analysis, 28-day mortality were analyzed using the same predictors including admission hypoxemia. Chi square test and regression model were built to calculate OR with 95%CI.

### Results

Among 383 patients (median age, 56 years [IQR, 24–110]; 49.9% men); 95 had neurological complications (cases) and 288 did not (controls). Development of neurological complications among COVID-19 related hospitalizations was significantly associated with old age >71 yrs. (cases, 23.2%; controls, 13.5%; OR, 3.31; 95% CI, 1.28–8.55), presence of

**Funding:** The authors extend their appreciation to the deanship of scientific research at King Khalid University, Abha, Saudi Arabia, for funding this work through research groups program under grant # RGP.1/227/43.

**Competing interests:** The authors have declared that no competing interests exist.

diabetes mellitus (37.9% vs. 24%; OR, 1.9; 95% CI, 1.2–3.1), admission hyperglycemia (BSR 351–600 mg/dl), (29.5% vs. 7.6%; OR, 3.11; 95%CI, 1.54–6.33), raised D-dimer levels 5000–10,000 ng/ml (41% vs. 11.8%; OR, 5.2; 95% CI, 3.02–8.9), prophylactic dose anticoagulation (43.2% vs. 28.1%; OR, 1.9; 95%CI, 1.2–3.1), and unvaccinated status of COVID-19 patients (90.5% vs. 75.6%; OR, 3.01; 95% CI, 1.44–6.25). Neurological complications with COVID-19 were associated with increased likelihood of death or invasive mechanical ventilation by day 28 (86.3% vs. 45.1%; OR, 7.66; 95% CI, 4.08–14.4). In case-only analysis (median age, 56 years [IQR, 27,110]; 50.5% women), 67 (70.5%) had CVE, 21 (22.1%) had Encephalitis, and 7 (7.4%) had GBS as neurological manifestations. 28-day mortality among these patients was strongly associated with a lower likelihood of vaccination. (6.1% cases vs. 30.8% controls; OR, .146; 95%CI, .033- .64), being younger 17–45 yrs. (12.2% vs. 46.2%; OR, .162; 95%CI, .045-.58), having no comorbid condition (19.5% vs. 61.5%; OR, .151; 95%CI, .044- .525), having cerebrovascular events and GBS as type of neurological manifestation (76.8% vs.30.8%; OR, 7.46; 95%CI, 2.06–26.96), (2.4% vs. 38.4%; OR, .04; 95%CI, .007- 0.24) respectively, and presence of hypoxemia at admission (91.5% vs. 15.4%; OR, 58.92; 95%CI, 10.83–320.67).

## Conclusion

Old age, presence of Diabetes Mellitus, unvaccinated status of patients, high BSR at admission, high D-dimers, and prophylactic dose anticoagulation were identifies as increased risk factors for developing serious neurological complications among COVID-19 patients. Neurological problems in COVID-19 patients raised death risk 7.6-fold. The most common neurological complication was cerebrovascular events, followed by encephalitis and GBS. Unvaccinated status, cerebrovascular events, and admission hypoxemia are associated with an increased likelihood of 28-day mortality among these patients.

## Introduction

Although COVID-19 is typified by respiratory symptoms and complications, neurological presentations of disease must not be underestimated. At least one subjective neurological symptom has been reported in nearly 36 to 85% of COVID-19 patients, highlighting the importance of the disease's neurological repercussions [1, 2]. Initially, headache, gustatory and olfactory dysfunctions and confusion were the most common general non-specific neurological symptoms reported by COVID-19 patients while major neurological problems such as ischemic and hemorrhagic stroke, cerebral (sinus) venous thrombosis, Encephalitis, Guillain–Barré syndrome and posterior reversible encephalopathy syndrome (PRES) had also been observed with severe cases [3, 4]. With the introduction of variants of concern (VOC), such as the delta variant (B.1.167.2) of SARS-COV-2, the disease's severity and complications had been escalated [5]. Patients with severe COVID-19 infection had a higher rate of neurological complications than those with moderate infection. Moreover, neurological complications may have poor impact on outcome of COVID-19 patients [6, 7].

Neuropathology is caused by a combination of immunological and hypoxia-induced damage superimposed by coagulation abnormalities [8]. ACE 2 receptors being the principal binding sites for SARS-COV-2, are found in neurons, oligodendrocytes, astrocytes, substantia

nigra, posterior cortex, brainstem and olfactory bulb. As a result, SARS-CoV-2 has the ability to infect neurons and glial cells throughout the brain [8, 9]. Blood-brain barrier(BBB) is broken by elevated cytokine levels, chemokine, and free radicals linked to activation of the immune system. Triggered inflammatory cascade can cause reactive astrogliosis [10]. Once the virus has crossed the BBB and entered the CNS, it will be difficult to eliminate it since the nervous system lacks substantial histocompatibility antigens and the immune response will be confined to cytotoxic T cells. The patient finally develops acute encephalitis, viral toxic encephalopathy, or acute cerebrovascular accidents (CVAs) [8, 10]. Infectious toxic encephalopathy (ITE) is a reversible brain dysfunction syndrome marked by cerebral edema caused by systemic toxemia and hypoxia, which can result in delirium and coma. In COVID-19 patients, virus-induced cytokine storm and coagulation abnormalities increase the likelihood of acute CVA [11].

There are several clinical, laboratory and radiological investigations that can predict the severity of disease and outcome in COVID-19 patients assigned to respiratory complications [12]. The vast majority of studies on neurological complications, including meta-analyses, focus on the kind of neurological complication and the outcome, while predictors of neurological sequelae are underexplored [13, 14], but there are hardly any relevant predictors for neurological sequelae. So, we designed an observational study at COVID-19 High Dependency Unit of tertiary care hospital to investigate possible predicting factors that may put COVID-19 patients at risk of developing serious neurological complications, as well as to observe any impact of complications and predictors on patients' outcome.

## Material and methods

A Case Control study approved by Institutional Review Board, Department of medical education, Qauid e Azam Medical College Bahawalpur, vide letter no.1283, and was carried out between July 15th, 2021 and December 15th 2021, delta period of SARS-COV-2 in Pakistan, to find out risk factors for fatal neurological complications in admitted COVID-19 patients and their impact on outcome. Study comprised adults aged 18 years and above for potential eligibility through daily review of hospital admissions and electronic record of COVID-19 High Dependency Unit and Intensive Care Units, at Sir, Sadiq Abbasi Hospital, Bahawalpur, a tertiary care hospital that received referrals of COVID-19 confirmed patients from all primary, secondary, and tertiary care hospitals of division, Bahawalpur, Punjab, Pakistan. An informed written consent was taken from patients or their escorts. The STROBE (Strengthening the Reporting of Observational Studies in Epidemiology) reporting guideline was followed in this study.

The cases were patients with a clinical syndrome consistent with acute COVID-19 and a positive PCR for SARS-CoV-2 (done by extraction and amplification method on Quant Studio 5 applied biosystem machine) or suggested HRCT who were hospitalized within 10 days of symptom onset and developed an acute onset severe neurological complication at admission or within 28 days of hospital stay. Neurological manifestations were defined as recent onset focal neurological deficit or altered sensorium gained after having symptoms of COVID-19 or laboratory confirmation of it. In addition to a full neurological evaluation by a neurologist, neuroimaging such as computed tomography, MRI brain, CSF analysis, and Nerve Conduction Studies were performed where indicated. The study excluded participants having a history of previous neurological illnesses that left them with residual weakness and those with partial or unknown vaccination status. Considering the study's focus on serious neurological manifestations (focal neurological deficit, arterial or venous stroke, encephalitis leading to decrease in consciousness, or acute demyelinating disorders), as a complication of COVID-19, mild

neurological symptoms e.g. loss of taste and smell, headache and facial nerve palsy were excluded. Controls were recruited from a pool of eligible COVID-19 confirmed participants who had been hospitalized within two weeks of the case's enrollment and matched in gender and duration by a 3:1 ratio. Matching for admission dates minimized the risk of lead time bias.

Age, co morbid illnesses, blood sugar level in mg/dl, D-Dimer level in ng/ml, type and dose of anticoagulation, and prior vaccination status were exposure predictor variables. Participants were categorized into three major groups based on their age. Co morbid conditions and their number in each participant were also noted, with a particular emphasis on uncontrolled comorbid factors for severe COVID-19. Random Blood Sugar in mg/dl was also noted at the time of admission; however, it changed as a result of hospital and disease-related stress, as well as intravenous steroid. On the basis of their blood sugar levels, the participants were divided into three categories: less than 200 mg/dl, 201–350 mg/dl, and 351 to 600 mg/dl. D-dimers were checked within 24 hours of hospitalization, prior to the initiation of anticoagulation, based on the results participants were divided into four categories: less than 500ng/ml, 501-5000ng/ml, 5001 to 10000 ng/ml and more than 10,000 ng/ml. The D-dimers were expressed in ng/ml FEU (Fibrinogen equivalent unit). Oral anticoagulants, Low Molecular Weight Heparin (LMWH), and unfractionated heparin were identified as anticoagulation types given during hospitalization or prior to it. On the basis of dosage, anticoagulation was categorized as either therapeutic or prophylaxis. Enoxaparin 40 mg daily for a body weight of < 100 kg, 60 mg daily for a body weight of more than or equal to 100 kg, and Rivaroxaban 10 mg daily were considered thromboprophylactic doses of anticoagulation. Enoxaparin 1 mg/kg body weight twice daily and Rivaroxaban 15 or 20 mg daily were considered therapeutic anticoagulation doses. Patients' vaccination status was determined by entering their National Identity numbers, which were obtained from their hospital electronic records, into a national database of vaccination program, along with the dates of their first and second doses of vaccination, and categorizing them as fully vaccinated or unvaccinated while excluding the partially vaccinated participants.

In the main analysis, the primary outcome measure was the development of neurological complications at admission or within 28 days of hospital stay. We also gathered data on the severity of COVID-19. These outcome measures were recorded until hospital discharge or 28 days following hospitalization, whichever came first. The major classification of illness severity was based on a dichotomous measure that divided between individuals who died or required invasive mechanical ventilation and those who did not. To determine severity, the patient's highest ordinal level within the first 28 days of hospitalization was employed. To categorize COVID-19 severity, we used a customized version of the World Health Organization COVID-19 Clinical Progression Scale, which goes from infection naïve (level 0) to infected albeit asymptomatic (level1) to death (level 9). To determine severity, the patient's highest ordinal level within the first 28 days of hospitalization was employed. The highest severity level observed could range from level 4 to level9, including hospitalized with no oxygen support (level4), with standard oxygen therapy (level5), with high-flow nasal cannula or noninvasive ventilator support (level6), with mechanical ventilation (level7), or with mechanical ventilation and additional organ assistance (hemodynamic support, extracorporeal membrane oxygenation, or new Renal Replacement Therapy (level 9)). According to scale level 4 were categorized as moderate disease, level 5 as severe disease and level 6,7 and 8 as critical illness. Patients with severity levels 6, 7, 8 and 9 required ICU care.

In case-only (subgroup) analysis, patients with neurological complications were analyzed. The frequency and percentage of each type of neurological complication were noted. Cases and controls were categorized as those who died and those who did not, using the same exposure variables (predictors) as in the primary analysis, with the addition of type of neurological

manifestation and hypoxemia at the time of admission. The primary outcome measures were illness progression to death within 28-days of hospital admission, survival leading to discharge from the hospital with residual weakness, or complete remission of neurological symptoms, whichever occurred first. The secondary outcome measures were also same as in primary analysis. Furthermore, the correlation between each type of neurological manifestation and predictor variables was evaluated independently.

In statistical analysis, median with inter quartile range (IQR) were taken for continuous variables and frequencies and percentages for categorical variables in both primary and subgroup analysis. The association between outcome and predictor variables was calculated in term of odds ratios (ORs) with their 95% CI. Odds Ratio for whom a 95% Confidence Interval crossing the null is considered insignificant. Chi Square test of independence and multinomial logistic regression model were created to calculate OR and significance values for dichotomous and multilevel dependent variables wherever applicable, keeping p value $< .05$ to be statistically significant. The following equation was used to determine the exposure effect for the prevention or occurrence of outcome: exposure predictability of outcome = (1- OR) 100%. $R2$ value is used to determine the correlation between each type of neurological problem and the predictor variable in subgroup analysis. Data was analyzed using version 26 of IBM SPSS statistics.

## Results

During study period, total 105 cases were enrolled, however, 10 patients were excluded, and the major cause of exclusion was previous history of neurological illness with residual weakness and undiagnosed neurological manifestation. A total of 383 analyzed populations [median age 56 yrs; IQR (24, 110), 49.9% males, median no. of comorbid conditions 1 IQR (0, 4)], comprised 95 cases and 288 gender and duration matched controls. Of the total studied population, 62.4% of participants were from the age group 46–70 yrs., and 33.7% of patients had at least one comorbid condition for severe OVID-19. Frequency and percentages for categorical variables and median with IQR for continuous variables among cases and controls are given in Table 1.

Development of neurological complication among COVID-19 patients was significantly associated with old age ≥71 yrs. [22 (23.2%) cases Vs. 39(13.5%) controls; OR, 3.31; 95%CI, 1.28–8.55; p < .05], having Diabetes Mellitus as a comorbid condition [36(37.9%) vs. 69(24%); OR, 1.9; 95%CI, 1.2–3.1; p < .05], high D-dimer level 5000–10000 ng/ml [39(41%) vs. 34 (11.8%); OR, 5.20; 95%CI, 3.02–8.9; p < .01], prophylactic dose of anticoagulation [41(43.2%) vs. 81(28.1%); OR, 1.9; 95%CI, 1.2–3.1; p < .05] and unvaccinated status of COVID-19 patients [86(90.5%) vs. 218(75.7%); OR, 3.01; 95%CI, 1.44–6.25; p < .01]. With increased likelihood of vaccination, there was decreased chance of neurological complications, [9(9.5%) vs. 70(18.3%); OR, .332; 95%CI, .16-.69, p < .001]. Estimated vaccine effectiveness against development of neurological complications was (1–0.33) ×100 = 66.9%. Association between exposure variables and development of neurological complications are given in Fig 1.

With increased likelihood of neurological complications there was decreased chances of having admission hypoxemia [18(18.9%) cases vs. 11(3.8%) controls; OR, .17 95% CI; .07-.37, p < .001]. When keeping the critical illness as reference category in multinomial regression analysis, 19(20%) vs. 11(3.8%) had moderate disease, (OR, 6.008; 95% CI, 2.668–13.529; p < .001), 30(31.6%) Vs. 117(40.6%) had severe disease (OR, .892; 95%CI, .531–1.497; p = .665). This means that patients with neurological complications are more likely to have moderate disease than those who do not.

**Table 1. Characteristics of case patients (with neurological complications) and control patients (without neurological complications).**

| Characteristics | No. (%) | |
| --- | --- | --- |
| | Cases (n = 95) | Controls (n = 288) |
| Age Median, IQR | 56(27,110) | 56(25,99) |
| Age groups | | |
| 25–45 yrs. | 16(16.8) | 67(23.3) |
| 46–70 yrs. | 57(60) | 182(63.2) |
| ≥71 yrs. | 22(23.2) | 39(13.5) |
| No. of comorbid conditions Median, IQR | 1(0,4) | 1(0,4) |
| Comorbid conditions | | |
| No risk factor | 24(25.3) | 87(30.2) |
| DM | 36(37.9) | 69(24) |
| Cardiovascular diseases | 27(28.4) | 108(37.5) |
| Pulmonary disease | 1(1.1) | 11(3.8) |
| Obesity | 0 | 1(0.3) |
| Old Age | 7(7.4) | 12(4.2) |
| Vaccination status | | |
| Vaccinated | 9(9.5) | 70(18.3) |
| Unvaccinated | 86(90.5) | 218(75.7) |
| BSR(mg/dl) Median, IQR | 218(80,578) | 174(98,340) |
| <200 | 37(38.9) | 206(71.5) |
| 201–350 | 30(31.6) | 82(28.5) |
| 351–600 | 19(20) | 0 |
| D-dimers(ng/ml) | 5900(280,13,000) | 2100(320,7900) |
| ≤500 | 9(9.5) | 12(4.2) |
| 501–5000 | 28(29.5) | 225(78.1) |
| 5000–10,000 | 39(41) | 34(11.8) |
| >10,000 | 19(20) | 17(5.9) |

Eighty three (86.3%) cases Vs. 130(45.1%) had disease progression to death or shifting to invasive ventilator support, OR 7.66, 95%CI (4.08, 14.4), RR 5.1 95%CI (2.93, 8.81) and p < .001. So, patients who had neurological complications had 660% more chances of dying than those without them.

In a case-only (subgroup) analysis, [median age 56 yrs., IQR (27,110), 48 (50.5%) were females, 67 (70.5%) patients developed Cerebral Vascular Events (CVE), seven (7.4%) had GBS (Guillain Barre Syndrome) confirmed by Nerve Conduction Study, and 21 (22.1%) had Encephalitis]. Among patients with CVE, 55 (82%) had ischemic infarct of arterial territory on Computed Tomography, six (9%) had hemorrhagic infarct, and six (9%) patients had cerebral venous thrombotic events, including three cases with cavernous venous sinus thrombosis. Association between the death outcome and the predictor variables is depicted in Table 2.

The frequency and percentage of categorical variables were determined independently for each type of neurological condition, as given in Table 3.

COVID-19 patients with complete vaccination status had a lower risk of developing cerebrovascular events, (OR, 146; 95%CI, .044-.479; p.01); however, no significant association was found for GBS and encephalitis, p>.05, in the multinomial regression model where the absence of neurological complications was kept as a reference category Fig 2. Depicts the relationship between various types of neurological complications, age groups, and D-dimer levels in terms of $R^2$.

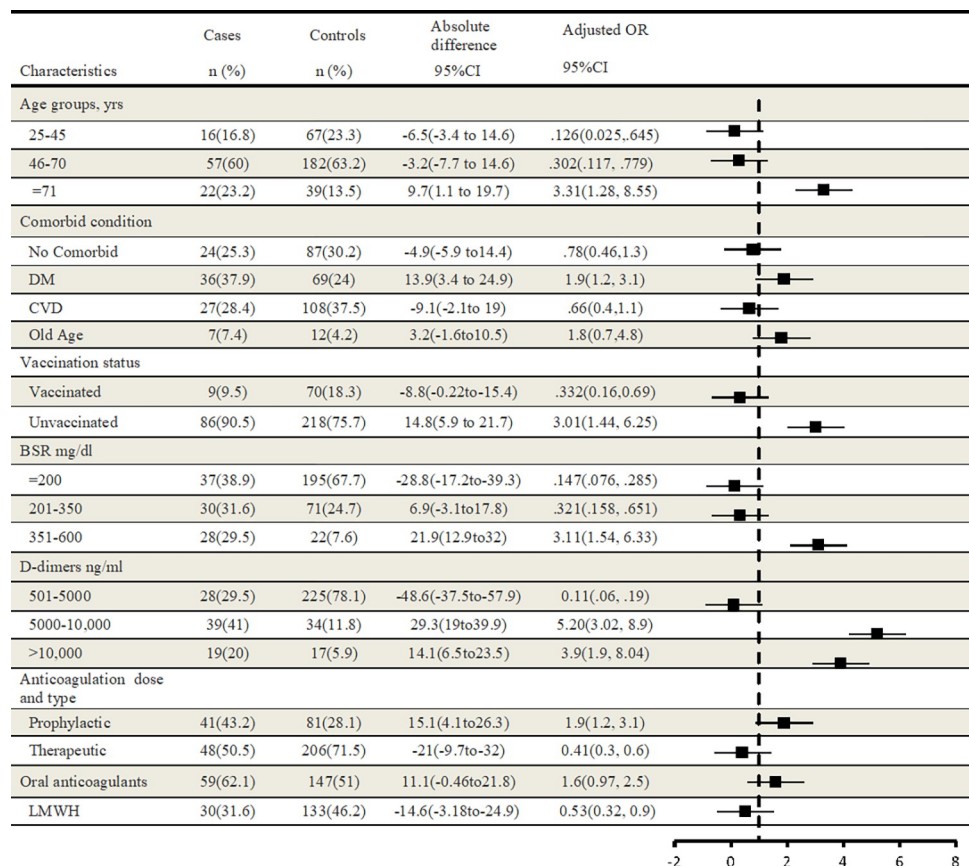

| Characteristics | Cases n (%) | Controls n (%) | Absolute difference 95%CI | Adjusted OR 95%CI | |
|---|---|---|---|---|---|
| **Age groups, yrs** | | | | | |
| 25-45 | 16(16.8) | 67(23.3) | -6.5(-3.4 to 14.6) | .126(0.025,.645) | |
| 46-70 | 57(60) | 182(63.2) | -3.2(-7.7 to 14.6) | .302(.117, .779) | |
| =71 | 22(23.2) | 39(13.5) | 9.7(1.1 to 19.7) | 3.31(1.28, 8.55) | |
| **Comorbid condition** | | | | | |
| No Comorbid | 24(25.3) | 87(30.2) | -4.9(-5.9 to14.4) | .78(0.46,1.3) | |
| DM | 36(37.9) | 69(24) | 13.9(3.4 to 24.9) | 1.9(1.2, 3.1) | |
| CVD | 27(28.4) | 108(37.5) | -9.1(-2.1to 19) | .66(0.4,1.1) | |
| Old Age | 7(7.4) | 12(4.2) | 3.2(-1.6to10.5) | 1.8(0.7,4.8) | |
| **Vaccination status** | | | | | |
| Vaccinated | 9(9.5) | 70(18.3) | -8.8(-0.22to-15.4) | .332(0.16,0.69) | |
| Unvaccinated | 86(90.5) | 218(75.7) | 14.8(5.9 to 21.7) | 3.01(1.44, 6.25) | |
| **BSR mg/dl** | | | | | |
| =200 | 37(38.9) | 195(67.7) | -28.8(-17.2to-39.3) | .147(.076, .285) | |
| 201-350 | 30(31.6) | 71(24.7) | 6.9(-3.1to17.8) | .321(.158, .651) | |
| 351-600 | 28(29.5) | 22(7.6) | 21.9(12.9to32) | 3.11(1.54, 6.33) | |
| **D-dimers ng/ml** | | | | | |
| 501-5000 | 28(29.5) | 225(78.1) | -48.6(-37.5to-57.9) | 0.11(.06, .19) | |
| 5000-10,000 | 39(41) | 34(11.8) | 29.3(19to39.9) | 5.20(3.02, 8.9) | |
| >10,000 | 19(20) | 17(5.9) | 14.1(6.5to23.5) | 3.9(1.9, 8.04) | |
| **Anticoagulation dose and type** | | | | | |
| Prophylactic | 41(43.2) | 81(28.1) | 15.1(4.1to26.3) | 1.9(1.2, 3.1) | |
| Therapeutic | 48(50.5) | 206(71.5) | -21(-9.7to-32) | 0.41(0.3, 0.6) | |
| Oral anticoagulants | 59(62.1) | 147(51) | 11.1(-0.46to21.8) | 1.6(0.97, 2.5) | |
| LMWH | 30(31.6) | 133(46.2) | -14.6(-3.18to-24.9) | 0.53(0.32, 0.9) | |

**Fig 1. Association between predictor variables and development of neurological complications among COVID-19 patients.** DM (Diabetese Mellitus), CVD (cardiovascular Diseases), BSR (blood sugar random), LMWH (Low molecular weight heparin). Association is measured in term of OR 95% CI shown in forest plot. In this model, an odds ratio <1 indicated decreased likelihood of neurological complication, and vice versa.

## Discussions

In this analysis of adults hospitalized for COVID-19 between mid-July 2021 and mid-December 2021 (delta period in Pakistan), old age, diabetes mellitus as a comorbid factor, unvaccinated status, admission hyperglycemia 351–600 mg/dl, raised D-dimer level 5000–10000 ng/ml, and prophylactic dose anticoagulation were associated with an increased risk of neurological complications among COVID-19 admitted patients. Young age, complete vaccination, BSR less than 350 mg/dl, and D-dimers less than 5000ng/ml, as well as therapeutic dosage anticoagulation, were associated with a lower risk of neurological problems among CVOID-19 admitted patients. COVID-19 patients with neurological problems are less likely to have hypoxemia at admission and more likely to have moderate lung disease compared to COVID-19 patients without neurological problems, yet they had a 660% increased risk of death compared to the other group. In the case-only (subgroup) analysis, being young (25–45 yrs.), not having a comorbid condition, being fully vaccinated, and having a D-dimers level below 5000 ng/mL were significantly associated with a lower risk of death, whereas being unvaccinated, having admission hypoxemia, and cerebrovascular event as a type of neurological complication were higher risk factors. Cerebrovascular events were the most prevalent form of neurological complication among cases, and full vaccination provides nearly 85% protection against their development in COVID-19 breakthrough cases who require hospitalization. The majority of patients who acquired GBS, were younger than those who had cerebrovascular events and encephalitis.

**Table 2. Association between predictor variables and death outcome among patients with neurological complications.**

| Characteristics | Deaths n(%) | Survivals n(%) | OR 95%CI | p value |
|---|---|---|---|---|
| Total no. | 82 | 13 | | |
| Age yrs, Median IQR | 59(30,110) | 46(27,67) | | |
| Age group | | | | |
| 25–45 yrs. | 10(12.2) | 6(46.2) | .162(.045 to .58) | < .01 |
| 46–70 yrs. | 50(61) | 7(53.8) | 1.34(.412 to 4.35) | >.05 |
| ≥71 yrs. | 22(26.8) | 0 | 10.04(.57, 176.02) | >.05 |
| No. of comorbid, IQR | 1(0,4) | 0(0,3) | | |
| No risk factor | 16(19.5) | 8(61.5) | .151(.044, .525) | .003 |
| DM | 34(41.5) | 2(15.4) | 3.895(.811, 18.72) | .09 |
| CVD | 24(29.3) | 3(23.1) | 1.379(.349, 5.46) | .65 |
| Pulmonary disease | 1(1.2) | 0 | .496(.019,12.84) | .7 |
| Old Age | 7(8.5) | 0 | 2.68(.144, 49.77) | .51 |
| Vaccination Status | | | | |
| Vaccinated | 5(6.1) | 4(30.8) | .146(.033, .64) | < .05 |
| Unvaccinated | 77(93.9) | 9(69.2) | 6.84(1.55, 30.22) | < .05 |
| Neurological manifestation | | | | |
| Cerebrovascular Accidents | 63(76.8) | 4(30.8) | 7.46(2.06, 26.96) | < .01 |
| GBS | 2(2.4) | 5(38.4) | .04(.007, 0.24) | < .001 |
| Encephalitis | 17(20.7) | 4(30.8) | .588(.16, 2.14) | .422 |
| D-dimers, median(IQR) | 6500 (280,13000) | 1500(350,7800) | | |
| ≤500 | 5(6.1) | 4(30.8) | .144(.033, .63) | .016 |
| 501–5000 | 21(25.6) | 7(53.8) | .29(.09,.97) | .04 |
| 5000–10,000 | 37(45.1) | 2(15.4) | 4.52(.94, 21.69) | .05 |
| >10,000 | 19(23.2) | 0 | 8.29(.47, 145.94) | .14 |
| BSR, IQR | 232(80,578) | 167(120,429) | | |
| <200 | 30(36.6) | 7(53.8) | .16(.02, .137) | .095 |
| 201–350 | 25(30.5) | 5(38.5) | .185(.02, 1.69) | .136 |
| 351–600 | 27(32.9) | 1(7.7) | 5.40(.590, 49.46) | .136 |
| Anticoagulation dose | | | | |
| No anticoagulant | 6(7.3) | 0(0) | | |
| Prophylactic | 34(41.5) | 7(53.8) | .620(.186, 2.06) | >.05 |
| Therapeutic | 42(51.2) | 6(46.2) | 1.612(.484,5.37) | >.05 |
| Hypoxemia at Admission | 75(91.5) | 2(15.4) | 58.92(10.83, 320.67) | < .001 |

D-dimer levels were found to be higher among patients with cerebrovascular events and lower in patients with GBS, but had no significant correlation with encephalitis.

OR in this analysis corresponds to increased risk of neurological complications in older age group of COVID-19 patients [1, 15]; Whereas in a previous study young age group is found to be risk factor for neurological complications. But the majority neurological manifestations were nonspecific in the study [16]. Those with Diabetes Mellitus as a comorbid factor with severe covid-19 were more likely to develop neurological sequelae, whereas patients with cardiovascular illness had no such risk [11].

Vaccination was less likely among participants with neurological complication. Estimated vaccine effectiveness against serious neurological sequelae among COVID-19 breakthrough infections is 66.9%, which is comparable to vaccine effectiveness against prevention of hospitalizations and disease progression to death among COVID-19 admitted patients for respiratory complications [17].

**Table 3. Type of neurological complications and categorical variables.**

| characteristics | Cerebrovascular events n(%) | Encephalitis n(%) | Guillain Barre Syndrome n(%) |
|---|---|---|---|
| Age groups | | | |
| 17–45 yrs. | 9/67(13.4) | 3/21(14.3) | 4/7(57.1) |
| 46–70 yrs. | 43/67(64.2) | 11/21(52.4) | 3/7(42.9) |
| ≥71 yrs. | 15/67(22.4) | 7/21(33.3) | 0/7(0) |
| No risk factor | 15/67(22.4) | 3/21(14.3) | 6/7(85.7) |
| DM | 25/67(37.3) | 10/21(47.6) | 1/7(14.3) |
| CVD | 20/67(29.9) | 7/21(33.3) | 0/7(0) |
| Vaccinated | 3/67(4.5) | 4/21(19) | 2/7(28.6) |
| Unvaccinated | 64/67(95.5) | 17/21(81) | 5/7(71.4) |
| BSR mg/dl | | | |
| ≤200 | 21/67(31.3) | 11/21(52.4) | 5/7(71.4) |
| 201–350 | 26/67(38.8) | 3/21(14.3) | 1/7(14.3) |
| 351–600 | 20/67(29.9) | 7/21(33.3) | 1/7(14.3) |
| D-dimers (ng/ml) | | | |
| ≤500 | 0/67(0) | 5/21(23.8) | 4/7(57.1) |
| 501–5000 | 10/67(14.9) | 15/21(71.4) | 3/7(42.9) |
| 5001–10,000 | 39/67(58.2) | 0/21(0) | 0/7(0) |
| >10,000 | 18/67(26.9) | 1/21(4.8) | 0/7(0) |
| Anticoagulation | | | |
| Prophylactic dose | 36/67(53.7) | 11/21(52.4) | 1/7(14.3) |
| Therapeutic dose | 26/67(38.8) | 10/21(47.6) | 6/7(85.7) |

Patients with severe hyperglycemia (>350 mg/dl) were 3.1 times more likely to suffer neurological complications, while patients with markedly raised D-dimers (> 5,000ng/µl) were 5.2 times more probable. Patients on prophylactic dose anticoagulation were 1.9 times more likely to develop neurological complications, while those on therapeutic dose anticoagulation had

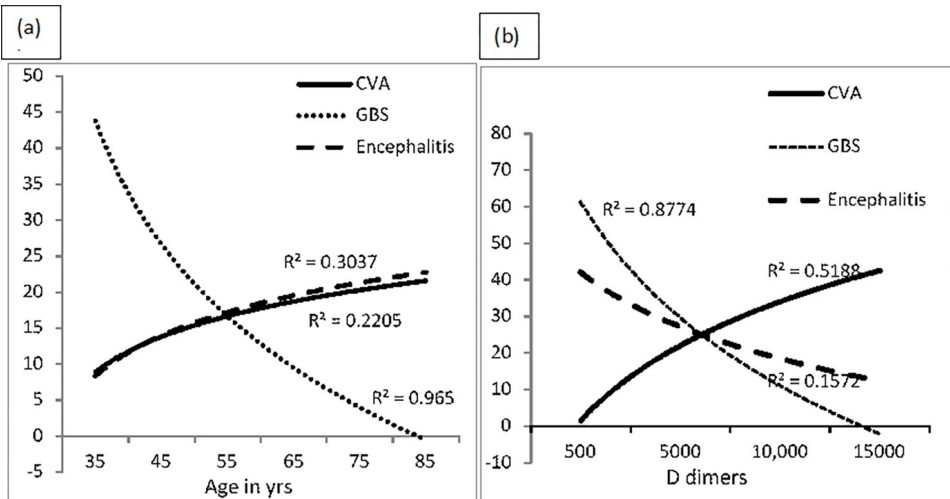

**Fig 2. Correlation between age in yrs. and D-dimer levels with different types of neurological manifestations.**
CVA (Cerebrovascular Events), GBS (Guillen Barre Syndrome). (a) Shows patients with GBS were from young age group, there was decline in cases with increase in age, and Cerebrovascular Events were more from old age group and same with encephalitis. (b) Shows relationship between D-dimers and neurological manifestation, patients with GBS had negative correlation with D-dimers whereas patients with CVA had positive correlation elaborated as $R^2$ value.

59% lower chances of developing the sequelae [18, 19]. There was no significant effect of therapeutic anticoagulation on death outcome among patients who already developed neurological complications [20]. In comparison to oral anticoagulation those participants who were on LMWH (enoxaparin) had 47% lesser chances of getting the neurological complication.

In COVID-19 patients who have neurological complications, the risk of disease progression to death is 7.6 times higher than in those who do not [11]. Almost 80% patients had severe respiratory disease at time of hospitalization. So, patients with severe COVID-19 are more likely to get neurological complications [21].

Among patients who developed neurological complications Cerebrovascular events (76.8%) were the most common presentation, followed by encephalitis and GBS [9, 22]. In this group, 18.9% of patients did not have any hypoxemia at the time of admission to the hospital, while 20% of patients had moderate respiratory complications, highlighting that neurological complications can occur as distinct and substantial consequence of COVID-19. When analyzing the risk factors for mortality and survival among patients with neurological complications, research indicates that young patients with no risk factors, who are fully vaccinated, have a low D-dimer, and do not have hypoxemia at the time of hospitalization are more likely to survive and have a better outcome. There was no significant association between the rest of the risk factors evaluated and death or survival outcomes in these patients. Mostly, 71.4% patients with GBS had good prognosis [23]. Whereas, there was great mortality among other type of complications i.e. Stroke and Encephalitis [6, 16].

## Conclusion

Old age, diabetes mellitus, unvaccinated status, admission hyperglycemia, raised D-dimer levels, and prophylactic dose anticoagulation were all risk factors for neurological problems in COVID-19 admitted patients. Neurological problems in COVID-19 patients raised death risk 7.6-fold. Young age, being fully vaccinated, having no comorbidity, and a low D-dimer level were substantially related to a lower chance of death, whereas being unvaccinated, having admission hypoxemia, and having a cerebrovascular event were greater risk factors. The most common fatal neurological complications among COVID-19 patients were cerebrovascular events, followed by encephalitis. Because neurological complications had an unexpectedly high mortality rate, this COVID-19-related complication should not be overlooked in hospitalized patients, and each patient should be thoroughly examined and checked for the presence of risk factors for neurological manifestations to prioritize them and consider any preventable consequences.

## Limitations of study

Retrospective nature of study and different types of neurological complications among cases are limitations of study. We need longitudinal prospective studies among COVID-19 patients in order to establish strength to observations. Moreover, study duration was short and only included delta period so the result cannot be generalized to all variants of SARS-COV-2.

## Supporting information

**S1 Dataset.**
(XLSX)

## Author Contributions

**Conceptualization:** Javaria Aslam, Shoaib Luqman.

**Data curation:** Javaria Aslam, Sadaf Nazly, Alina Saeed, Muhammad Sohail Tariq.

**Formal analysis:** Javaria Aslam, Alina Saeed, Irrum Aneela.

**Funding acquisition:** Sultan Yahya Mohammad Alfaife.

**Methodology:** Javaria Aslam, Muhammad Sohail Tariq, Sultan Yahya Mohammad Alfaife, Irrum Aneela.

**Software:** Irrum Aneela.

**Supervision:** Irrum Aneela.

**Validation:** Irrum Aneela.

**Visualization:** Irrum Aneela.

**Writing – original draft:** Javaria Aslam.

**Writing – review & editing:** Javaria Aslam, Shoaib Luqman, Sultan Yahya Mohammad Alfaife.

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
