## [Decision Letter · Decision Letter 0]

14 Jul 2022

PONE-D-22-05696Predictors of Fatal Neurological Complications among Admitted COVID-19 Patients with their Implication in Outcome: A Case Control StudyPLOS ONE

Dear Dr. Aslam,

Thank you for submitting your manuscript to PLOS ONE. After careful consideration, we feel that it has merit but does not fully meet PLOS ONE’s publication criteria as it currently stands. Therefore, we invite you to submit a revised version of the manuscript that addresses the points raised during the review process.

We look forward to receiving your revised manuscript.

Kind regards,

Soham Bandyopadhyay

Academic Editor

PLOS ONE

Journal Requirements:

a) Did participants provide their written or verbal informed consent to participate in this study?

5. Please remove your figures from within your manuscript file, leaving only the individual TIFF/EPS image files, uploaded separately.  These will be automatically included in the reviewers’ PDF.

Reviewers' comments:

Reviewer's Responses to Questions

**Comments to the Author**

1. Is the manuscript technically sound, and do the data support the conclusions?

Reviewer #1: Yes

Reviewer #2: No

2. Has the statistical analysis been performed appropriately and rigorously? 

Reviewer #1: Yes

Reviewer #2: Yes

3. Have the authors made all data underlying the findings in their manuscript fully available?

Reviewer #1: Yes

Reviewer #2: Yes

4. Is the manuscript presented in an intelligible fashion and written in standard English?

Reviewer #1: Yes

Reviewer #2: No

5. Review Comments to the Author

Reviewer #1: Keywords should include important terms like Cerebrovascular events, and COVID 19. The keywords used are general, non-specific, and should be edited.

When mentioning reference numbers in the parenthesis, a full stop should be following the references at end of the sentence, not preceding the references.

Line 78-82: Is the author trying to suggest an increased risk of neurological disease with Delta variant as compared to COVID 19 in general? Please clarify for the readers.

Please abstain from a colloquial style of wiring that will also remove redundancy from the text.

Instead of mentioning molecular testing for COVID, please mention the detail with the type of PCR and instrumentation used for testing.

Line 150 - Were partially vaccinated patients included or excluded from the study? It should be clarified.

Line 192 - 18 patients among the cases had no hypoxemia at the time of admission, please state what was the reason for admission. Neurological symptoms or any other criteria used for admission should be mentioned.

Line 197-198 It will be interesting to mention respiratory illness severity and type of respiratory support in relation to neurological complications as increased risk of dying from covid 19.

Reviewer #2: This article requires an overhaul to address major concerns with formatting, stylistic approaches, grammar, methodology, and the presentation of your findings. While significant as a study from this corner of the world, it requires a more scientific approach towards your statistical inclusion of predictors, case-matching strategy, and extrapolation of findings. I believe with improvements and revisiting your data, this study can become a significant and worthy publication.

6. PLOS authors have the option to publish the peer review history of their article (what does this mean?). If published, this will include your full peer review and any attached files.

Reviewer #1: **Yes: **Bushra Tehreem

Reviewer #2: No

---

## [Author Response · Author response to Decision Letter 0]

2 Aug 2022

Response to Academic editor:

We much appreciate your consideration of this paper for publication. We have meticulously rectified the shortcomings identified by the academic editor. 

1. The manuscript has been modified to adhere to the journal's specifications.

2. The revised manuscript includes a clear statement that written consent was obtained from study participants.

3. Dataset has been given in supporting information in xlsx file.

4. Full ethic statement is included in the methodology section of revised manuscript.

5. Figures are withdrawn from the manuscript file and converted to TIFF format in separate files.

Response to Reviewer #1:

Thank you very much for your valuable insight into our work. We have addressed your comments carefully

1. In the revised manuscript, Key words have been edited and important terms such as cerebrovascular events, COVID-19, encephalitis, predictors, and outcome have been added.

2. Full stops are correctly placed following the reference number at the end of sentences.

3. The study was conducted during the delta period of SARS-CoV-2 in Pakistan, which was the worst in terms of disease spread, severity, and fatality. Moreover, more neurological problems were reported compared to earlier waves. Therefore, it was important to highlight its fatality in the introduction, despite the fact that it was not compared to the other waves and, as stated in the study's limitations, its results cannot be generalized.

4. Informal writing style has been avoided as much as possible in the revised version.

5. Type and specifications of SARS-COV-2 PCR testing have been described in detail in the revised manuscript's methodology section.

6. Only fully vaccinated or unvaccinated participants were included in the study, excluding partially vaccinated patients from both cases and controls. A clear statement on vaccination status is included in both the methodology and results sections of revised manuscript.

7. The research was conducted at a hospital that received COVID-19 confirmed cases from other tertiary care hospitals. In the setting of a pandemic, the hospital received all sick admissions with positive PCR or HRCT findings diagnostic of COVID pneumonia with acute onset symptoms. Eighteen patients did not have hypoxemia at the time of hospitalization, but had positive PCR for SARS-COV-2 and extra pulmonary symptoms such as neurological or HRCT findings suggestive of COVID pneumonia without hypoxemia.

8. All patients with neurological complications were evaluated for the severity of respiratory illness, termed "Disease severity," which was measured by the WHO COVID-19 clinical progression scale in the methodology section. According to the scale, patients with moderate disease had no hypoxemia and did not require any oxygen support, patients with severe disease had hypoxemia and required oxygen support via simple face mask or rebreathing mask, and patients with critical illness required noninvasive or invasive ventilator support and required ICU care in general.

Response to Reviewer #2:

Thank you very much for your valuable comments for our original work. We have carefully put an effort to revise all sections of the manuscript to satisfy the reviewer's major concerns. Errors in grammar and sentence structure have been addressed thoroughly. Red highlighting denotes alterations to the manuscript.

Abstract:

Background in the abstract section has been edited for grammatical errors and extended to make it more comprehensive. Result section of structured abstract has been revised, clearly explaining the OR in scientific way. Conclusion of abstract also revised to conclude the results in a meaningful and scientific way.

Introduction:

Highlighted sentence has been edited with required changes and two more refernces on # 13 and 14 are added.

Methadology:

The methodology section has been thoroughly revised while preserving the original methodology. In the first paragraph of the methodology, the duration, design, and location of the study, as well as the ethical statement and design principles, are outlined in detail. The following paragraph describes the participants who defined the cases and matched controls. In the following paragraphs, predictor variables and primary and secondary outcome measures are discussed in detail in primary and subgroup analysis. In the last statistical analysis is mentioned. In order to make the methodology section more comprehensible for the readers, the majority of the text is rearranged and modified with pertinent amendments.

1. Division name is mentioned as Bahawalpur, Punjab, Pakistan.

2. Mild and serious neurological complications are described with example in the section where cases, controls and, inclusion and exclusion criteria are described.

3. Hospital admission criteria was positive PCR or HRCT chest suggestive of COVID-19 pneumonia and its related complications e.g high grade persistent fever, hypoxemia needing respiratory assistance, serious neurological complications , cardiac complications like myocarditis or renal complications like acute kidney injury.

4. In order to avoid lead time bias, controls are recruited within two weeks after case enrollment. Since COVID-19 presentation is with a brief history of febrile illness and pulmonary and extra pulmonary complications, it has an abrupt onset. While defining the potential study participants, it was stated that they must be admitted within 10 days after symptom onset or a positive PCR result. Obtaining a patient's PCR testing history was a second method for excluding participants who had persistent positive test. Covid-19.pshealthpunjab.gov.pk, provides a record of each positive or negative test conducted on the individual based on national Identity number. Patients with long history of symptom onset or positive PCR test, for more than 10 days were excluded. 

Results:

The results section has been revised by revisiting the data and rearranging the results. With the aid of text, table 1 and figure 1, the main analysis’ results are outlined in detail. Afterwards, a case-only (subgroup) analysis is presented using text and table 2. In the last section, different neurological manifestations were evaluated independently in relation to predictor variables and outcome variables. To display the results of this investigation in a scientific manner, fig. 2 and a new table 3. Were incorporated. 

While revisiting and presenting the data in a more effective manner, its originality was kept intact and no changes were made to the original data or statistics.

The discussion section has been meticulously revised to eliminate all grammatical mistakes. The first paragraph is substantially modified by incorporating additional results' conclusions.

The conclusion section is rewritten to better summarize the study's findings and provide valuable suggestions.

---

## [Decision Letter · Decision Letter 1]

30 Aug 2022

Predictors of Fatal Neurological Complications among Admitted COVID-19 Patients with their Implication in Outcome: A Case Control Study

PONE-D-22-05696R1

Dear Dr.  Aslam

We’re pleased to inform you that your manuscript has been judged scientifically suitable for publication and will be formally accepted for publication once it meets all outstanding technical requirements.

Kind regards,

Soham Bandyopadhyay

Academic Editor

PLOS ONE

Reviewers' comments:

Reviewer's Responses to Questions

**Comments to the Author**

1. If the authors have adequately addressed your comments raised in a previous round of review and you feel that this manuscript is now acceptable for publication, you may indicate that here to bypass the “Comments to the Author” section, enter your conflict of interest statement in the “Confidential to Editor” section, and submit your "Accept" recommendation.

Reviewer #2: All comments have been addressed

2. Is the manuscript technically sound, and do the data support the conclusions?

Reviewer #2: Yes

3. Has the statistical analysis been performed appropriately and rigorously? 

Reviewer #2: Yes

4. Have the authors made all data underlying the findings in their manuscript fully available?

Reviewer #2: Yes

5. Is the manuscript presented in an intelligible fashion and written in standard English?

Reviewer #2: Yes

6. Review Comments to the Author

Reviewer #2: This revision is impressive and merits publication, given the authors' commitment to rigorous science, statistical method, and presentation of their findings. No further changes are recommended at the present moment.

7. PLOS authors have the option to publish the peer review history of their article (what does this mean?). If published, this will include your full peer review and any attached files.

Reviewer #2: **Yes: **M. Hamza Bajwa

---

## [Editor Report · Acceptance letter]

15 Sep 2022

PONE-D-22-05696R1 

Predictors of Fatal Neurological Complications among Admitted COVID-19 Patients with Their Implication in Outcome: A Case Control Study 

Dear Dr. Aslam:

I'm pleased to inform you that your manuscript has been deemed suitable for publication in PLOS ONE. Congratulations! Your manuscript is now with our production department. 

Kind regards, 

on behalf of

Dr. Soham Bandyopadhyay 

Academic Editor

PLOS ONE